# GENERATIVE QUESTION ANSWERING: LEARNING TO ANSWER THE WHOLE QUESTION

**Mike Lewis & Angela Fan**
Facebook AI Research
{mikelewis,angelafan}@fb.com

## ABSTRACT

Discriminative question answering models can overfit to superficial biases in datasets, because their loss function saturates when any clue makes the answer likely. We introduce generative models of the joint distribution of questions and answers, which are trained to explain the whole question, not just to answer it. Our question answering (QA) model is implemented by learning a prior over answers, and a conditional language model to generate the question given the answer—allowing scalable and interpretable many-hop reasoning as the question is generated word-by-word. Our model achieves competitive performance with comparable discriminative models on the SQUAD and CLEVR benchmarks, indicating that it is a more general architecture for language understanding and reasoning than previous work. The model greatly improves generalisation both from biased training data and to adversarial testing data, achieving state-of-the-art results on ADVERSARIALSQUAD.

## 1 INTRODUCTION

Question answering tasks are widely used for training and testing machine comprehension and reasoning (Rajpurkar et al., 2016; Joshi et al., 2017). However, high performance has been achieved with only superficial understanding, as models exploit simple correlations in the data (Weissenborn et al., 2017; Zhou et al., 2015). For example, in Visual QA (Agrawal et al., 2017), the answer to *What colour is the grass?* can be memorised as *green* without considering the image (Figure 1).

We argue that this over-fitting to biases is partly caused by discriminative loss functions, which saturate when simple correlations allow the question to be answered confidently, leaving no incentive for further learning on the example.

We propose generative QA models, using Bayes' rule to reparameterise the distribution of answers given questions in terms of the distribution of questions given answers. We learn a prior over answers and a conditional language model for generating the question—reducing question answering to sequence-to-sequence learning (Sutskever et al., 2014), and allowing many-hop reasoning as the model explains the whole question word-by-word.

Generative loss functions train the model to explain all question words, even if the answer is obvious. For example, a model cannot assign high probability to generating the question *What colour is the grass?* without learning a dependency between the image and the word $grass$. We show that this method allows much improved generalisation from biased training data and to adversarial test data, compared to state-of-the-art discriminative models.

Word-by-word generative modelling of questions also supports chains of reasoning, as each subpart of the question is explained in turn. Existing methods use a pre-specified number of reasoning steps (Sukhbaatar et al., 2015; Hudson & Manning, 2018), which may be too many steps on easy cases, and too few on long and complex questions. We instead perform an interpretable reasoning step for each question word, and achieve 97.7% accuracy on the CLEVR benchmark (Johnson et al., 2017).

Our approach opens a promising new direction for question answering, with strong results in language understanding, reasoning and generalisation.

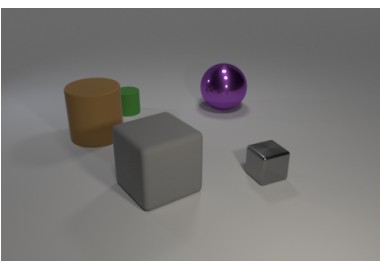

*Is the purple thing the same shape as the large gray rubber thing?*
*Does the green rubber object have the same shape as the gray thing that is on the right side of the big purple object?*

Whilst filming in Mexico City, speculation in the media claimed that the script had been altered to accommodate the demands of Mexican authorities reportedly influencing details of the scene and characters, casting choices, and modifying the script in order to portray the country in a "positive light" in order to secure tax concessions and financial support worth up to $20 million for the film. This was denied by producer Michael G. Wilson.

*Which Bond producer would not confirm that the film had been changed to accommodate Mexican authorities?*

(a) Two CLEVR questions. Both can be answered *no* using only subsets of the available information. A generative model must learn to perform additional reasoning to assign high likelihood to the complete question-answer pair. Word-by-word question generation allows a reasoning step to explain each word.

(b) A SQUAD question. A discriminative model can identify the only *producer*, and ignore the rest of the question. To generate the question and answer, our model needs coreference, negation and paraphrasing. These reasoning skills can improve generalisation on test examples with multiple plausible answers.

Figure 1: Examples of questions that can be answered using only some question words (underlined).

## 2 MODEL

### 2.1 OVERVIEW

We assume a dataset of examples with a question $q = q_{0..T}$, answer $a$, and context $c$ (in our experiments, $c$ is a document or image, but alternatives such as knowledge graphs could be used).

We train models to minimize the negative log likelihood of the joint distribution of questions and answers given the context, $-\log p(q, a|c)$, which we decompose using the chain rule as:

$$\mathcal{L} = -\log p(a|c) - \sum_t \log p(q_t|a, c, q_{0..t-1}) \tag{1}$$

First $c$ is encoded, using a recurrent model for text (§2.2) and a using a convolutional model for images (§2.3).

Then, a prior over answers $p(a|c)$ is evaluated by scoring all possible answers (2.4).

The likelihood of the question $p(q|a, c)$ is modelled using a conditional language model (§2.5).

At test time, the answer maximizing $p(q, a|c)$ is returned[1] (§2.7).

Hyperparameters and training details are fully described in Appendix A.

### 2.2 DOCUMENT ENCODER

**Answer-independent Context Representation** We use contextualised word representations, similar to ELMo (Peters et al., 2018). We use character-based word embeddings (Kim et al., 2016) and train a 2-layer LSTM language model in the forward and reverse directions, using the WikiText-103 corpus (Merity et al., 2016) for domain-specificity[2]. These parameters are frozen, and not fine-tuned. Contextualised word representations are computed as a fully connected layer applied to the concatenation of the word embedding and the hidden states of each layer of the language model. We sum this representation with a trainable vector of size $d$ if the word occurs in the article title, giving the encoder information about the article topic. We follow this with bidirectional LSTMs of size $d/2$, concatenating the output in each direction and adding residual connections after each layer.

---

[1] $\text{argmax}_a\, p(q, a|c) = \text{argmax}_a\, p(a|q, c)$, so this is the most likely answer given the question.
[2] The original ELMo implementation was trained on GBW (Chelba et al., 2014) and fine-tuned on SQUAD.

What we now call gravity was not identified as a universal force until the work of Isaac Newton. [...] Galileo was instrumental in describing the characteristics of falling objects [...] this acceleration due to gravity towards the surface of the Earth is usually designated as and has a magnitude of about 9.81 meters per second squared [...], and points toward the center of the Earth. [...] **Distractor:** Object falls about 5 times faster on Mars.

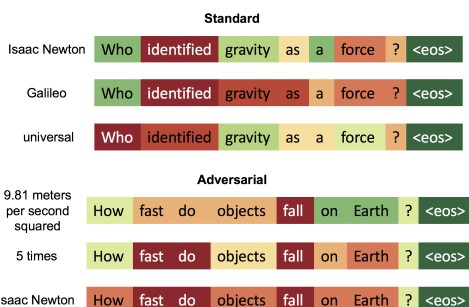

Figure 2: Probabilities of generating question words given different answers for a standard and an adversarial SQUAD question, allowing us to interpret which questions words are explained by the answer. In the standard setting, the model places greater probability on the question words that appear near *Isaac Newton*, such as *force* compared to *Galileo*. In the adversarial setting, the question word *Earth* distinguishes the true answer from the distractor.

**Answer Encoder** We assume answers $a$ are a span $i..j$ of the document. We represent the answer as a weighted sum of words within the span. For each word representation $\sum_{k=i..j} \sigma(w \cdot \tilde{c}_k)\tilde{c}_k$, where $w \in R^d$ is a trainable vector, and $\tilde{c}_k$ is the $k$th word in the answer-independent document representation. In contrast to previously proposed span representations (Lee et al., 2016; He et al., 2018), this approach allows the model to select arbitrarily many head words from the span.

**Answer-dependent Context Representation** Generative training makes it feasible to model more complex interactions between the answer and context than discriminative training, because only the correct answer is used. On SQUAD, we compute an answer-dependent document representation.

Here, we take the output of the answer-independent representation of each context word, and concatenate it with 32-dimensional embeddings of: a binary feature for whether the word is contained in the answer, its position relative to the answer start, and its position relative to the answer end. We also concatenate the element-wise product of the word representation and the answer encoding. We feed the result into 3 further layers of residual bidirectional LSTMs of size $d/2$.

## 2.3 IMAGE ENCODER

We use a simple image encoder, leaving reasoning to the question decoder.

Following Johnson et al. (2017), we take pre-trained features from the conv4 layer ResNet-101 model (He et al., 2016), giving 1024-dimensional features for a 14x14 image grid. We apply dropout, and project these representations to size $d$ using a 1x1 convolution, followed by batch normalisation (Ioffe & Szegedy, 2015) and a ReLU activation (Nair & Hinton, 2010). We then use 2 blocks of 3x3 convolutions, batch normalisation and ReLUs, and concatenate the final representation with a 32-dimensional positional encoding.

## 2.4 ANSWER PRIOR

We found modelling $p(a|c)$, the distribution over answers given the context, to be straightforward.

On SQUAD, we concatenate the start and end representations of the answer-independent context representation, combine them with a single hidden layer of size $2d$ and ReLU activation, and project this representation down to a score $s^{\text{endpoints}}(a, c)$. We also add an additional score based only on the length of the answer $s^{\text{length}}(a)$. Finally, we calculate: $p(a|c) = \frac{\exp\left(s^{\text{endpoints}}(a,c) + s^{\text{length}}(a)\right)}{\sum_{a'} \exp\left(s^{\text{endpoints}}(a',c) + s^{\text{length}}(a')\right)}$

On CLEVR, we simply apply a fully connected layer of size $2d$ and ReLU activation to the image representation, followed by a projection to the space of all possible answers and a softmax.

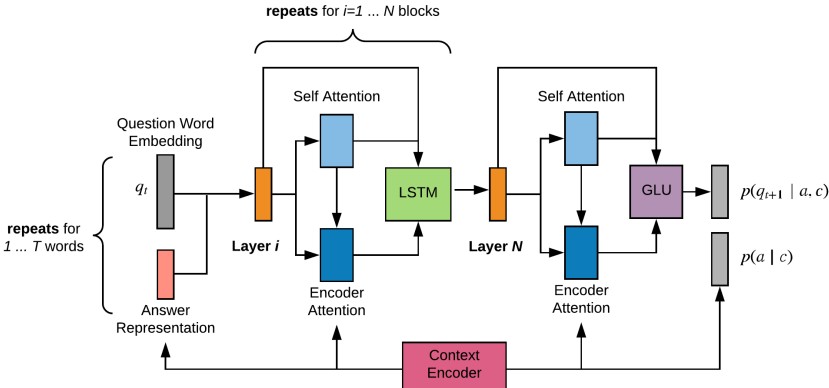

Figure 3: Architecture of GQA decoder. Multiple inputs to a layer indicates concatenation. Blocks after the first are connected with an additional residual connection, and LSTM cells also receive their state at time $t-1$ as an input.

## 2.5 QUESTION DECODER

We generate question words left-to-right with teacher forcing. We first embed words independently of the context (§2.5.1), then use a multi-layer RNN with attention to model interactions between the question and context (§2.5.2), and finally compute the likelihood of the next word (§2.5.3).

### 2.5.1 INPUT WORD EMBEDDINGS

To represent SQUAD question words independently of the answer and document, we use a pre-trained left-to-right language model (which can be viewed as a uni-directional version of ELMo), followed by a trainable LSTM layer of size $d$. On CLEVR, we simply train embeddings of size $d$.

### 2.5.2 DECODER BLOCKS

Decoder blocks are composed of a question self-attention layer (Vaswani et al., 2017) and a question-to-context attention mechanism, which are combined and fed into an LSTM (Figure 3). The context attention query is computed using both the previous layer state and the self-attention value. Blocks after the first are connected with residual connections.

Our attention mechanisms are implemented as in Vaswani et al., except that we use a single-headed attention, and a bias term $b_j$ is added to the query-key score for context position $j$. This bias term (calculated as a dot product of a shared trainable vector and context encoding $c_j$) allows the model to easily filter out parts of the context which are irrelevant to the question.

A final block again uses self-attention and question-to-document attention, which are then combined with a Gated Linear Unit layer (Dauphin et al., 2017) to output a vector of size $d$. The GLU layer uses a gating mechanism to select the relevant information for predicting the subsequent word.

### 2.5.3 OUTPUT WORD PROBABILITIES

Rare words are more challenging for generative than discriminative models, because it is easier to guess the meaning of a rare word from morphological clues than to generate it. SQUAD contains many rare words, because of the specialised vocabulary in Wikipedia. We improve modelling of rare words using a combination of an approximate character-based softmax and copy mechanism.

**Word Softmax** The simplest approach is to learn an embedding per word, replacing infrequent words with an *unknown* token. This approach cannot discriminate between different rare words. We use this method on CLEVR, which has a small vocabulary.

Figure 4: Final layer attention maps and word probabilities during question generation on a CLEVR validation question, when predicting the highlighted word. **(1)** The model considers all the rubber objects for predicting the next word. **(2)** Objects to the left of a rubber object are considered. **(3)** It describes the brown cylinder. **(4, 5)** Word distributions show the model understands the next words, but interestingly its attention focuses on the set of two objects meeting the constraints. In all cases, the word probability distributions are heavily skewed towards semantically valid choices.

**Character Softmax**   While there is an infinite space of possible output words, we approximate it by only normalising over the union of a list of frequent words, and any additional word that appears in any question or any document in the batch. We build a character-based representation for each candidate word using the pre-trained character CNN from §2.2, and add a trainable linear projection to size $d$. We combine character-based and word-based representations by summation.

**Pointer Mechanism**   We use a pointer mechanism (Vinyals et al., 2015) to improve the likelihood of the specialised vocabulary present in SQUAD, by copying article words. From the final hidden state $h_t$, the model first chooses whether to copy using a simple classifier $p^{\text{copy}}(h) = \sigma(w^{\text{copy}} \cdot h)$, where $w^{\text{copy}}$ is a trainable vector of size $d$. Then, the model interpolates between generating a word with softmax $p^{\text{gen}}(h)$ and copying context word $c_i$ using the question-to-context attention probability from the final layer $\alpha_t^i$: $p(q_t|q_{0:t-1}, c, a) = p^{\text{copy}}(h_t) \sum_i \alpha_t^i \mathbb{1}_{c_i=q_t} + (1 - p^{\text{copy}}(h_t))p^{\text{gen}}(q_t|h_t)$

## 2.6   FINE TUNING

Generative training using teacher forcing means that the model is not exposed to negative combinations of questions and answers at training time, so performance on these combinations may be weak when used for inference. For example, the model can overfit as a language model on questions, and ignore dependencies to the answer. Results can be improved by fine-tuning the model to make the question more likely under the gold answer than other plausible answers. Here, we minimize $- \log \frac{p(q|a,c)p(a|c)}{\sum_{a' \in A} p(q|a',c)p(a'|c)}$, where $A$ is the most likely 100 answer candidates from $p(a|c)$. The model performs poorly when trained using only this loss function, suggesting that generative pretraining allows the model to establish complex dependencies between the input and output, which can then be calibrated discriminatively (Table 2).

## 2.7   INFERENCE

We return the answer $a^*$ maximizing $a^* = \text{argmax}_a\, p(q|a, c)p(a|c)$, which requires evaluating the likelihood of the given question under each possible answer.

To efficiently handle the large number of possible answers in SQUAD, we use beam search. We evaluate $p(a|c)$ for all possible answer spans up to length 30, take the top 250 candidates, and only evaluate $p(q|a, c)$ for each of these. A correct answer is contained in the beam for over 98.5% of validation questions, suggesting that approximate inference is not a major cause of errors.

| Single Model | Development | | Test | |
|---|---|---|---|---|
| | EM | F1 | EM | F1 |
| RaSOR (Lee et al., 2016) | 66.4 | 74.9 | 67.4 | 75.5 |
| BiDAF (Seo et al., 2016) | 67.7 | 77.3 | 68.0 | 77.3 |
| DrQA (Chen et al., 2017) | 69.5 | 78.8 | 70.7 | 79.3 |
| R-Net (Wang et al., 2017) | 71.1 | 79.5 | 72.3 | 80.7 |
| Weaver (Raison et al., 2018) | 74.1 | 82.4 | 74.4 | 82.8 |
| DCN+ (Xiong et al., 2017) | 74.5 | 83.1 | 75.1 | 83.1 |
| QANet + data augmentation x3 (Yu et al., 2018) | 75.1 | 83.8 | 76.2 | 84.6 |
| BiDAF + Self Attention + ELMo (Peters et al., 2018) | - | 85.6 | 78.6 | 85.8 |
| Reinforced Mnemonic Reader (Hu et al., 2018) | 78.9 | 86.3 | 79.5 | 86.6 |
| GQA | 76.8 | 83.7 | 77.1 | 83.9 |

Table 1: Exact Match (EM) and F1 on SQuAD, comparing to the best published single models at the time of submission (September 2018).

| Single Model | Exact Match | F1 |
|---|---|---|
| GQA | 76.8 | 83.7 |
| GQA (no fine-tuning) | 72.3 | 80.1 |
| GQA (no generative training) | 64.5 | 72.2 |
| GQA (no character-based softmax) | 74.3 | 81.4 |
| GQA (no pointer mechanism) | 71.9 | 79.7 |
| GQA (no answer-dependent context representation) | 72.2 | 79.7 |
| GQA (answer prior only) | 13.4 | 16.1 |

Table 2: Development results on SQuAD for model ablations.

# 3 EXPERIMENTS

## 3.1 LARGE-SCALE READING COMPREHENSION

We evaluate our model (GQA) on the SQuAD dataset to test its robustness to diverse syntactic and lexical inferences. Results are shown in Table 1, and are competitive with comparable discriminative models, despite several years of incremental progress on discriminative architectures for this task. These results show the potential of generative models for such tasks.

Higher results have been reported using techniques such as ensembles, data augmentation, reinforcement learning with the end-task metric a reward, and breakthroughs in unsupervised pre-training.

Table 2 shows several ablations. It demonstrates the importance of the character-based softmax and pointer mechanism for modelling rare words, the need to model interactions between the answer and context, and a large improvement from fine-tuning the model with negative question-answer pairs.

The ablations also highlight that while fine-tuning the model with a discriminative objective substantially improves the results, performance is weak when trained discriminatively from scratch. This result suggests that generative training is learning additional relationships, but can benefit from being calibrated and exposed to negative question-answer pairs during fine tuning.

Table 3 shows an ablation study for the number of answer candidates considered in the beam of possible answers at inference time. Considering a larger number of answer candidates improves results, but increases the computational cost as the likelihood of the question must be calculated for each candidate.

## 3.2 MULTIHOP REASONING

We evaluate the ability of our model to perform multihop reasoning on the CLEVR dataset, which consists of images paired with automatically generated questions involving that test visual reasoning.

Table 4 shows that GQA achieves an accuracy of 97.7%, compared to 76.6% for a standard visual QA model, demonstrating that our generative architecture can perform complex reasoning. Integrat-

| # Answer Candidates | Exact Match | F1 |
|---|---|---|
| 250 | 76.8 | 83.7 |
| 200 | 76.6 | 83.4 |
| 100 | 76.2 | 83.1 |
| 50 | 74.6 | 81.4 |
| 10 | 55.7 | 61.4 |

Table 3: Development results on SQUAD, varying the beam size during inference.

| Single Model | Overall | Count | Exist | Compare Numbers | Query Attribute | Compare Attribute |
|---|---|---|---|---|---|---|
| Human | 92.6 | 86.7 | 96.6 | 86.5 | 95.0 | 96.0 |
| CNN+LSTM | 52.3 | 43.7 | 65.2 | 67.1 | 49.3 | 53.0 |
| CNN+LSTM+SA | 76.6 | 64.4 | 82.7 | 77.4 | 82.6 | 75.4 |
| CNN+LSTM+RN | 95.5 | 90.1 | 97.8 | 93.6 | 97.9 | 97.1 |
| CNN+GRU+FiLM | 97.6 | 94.3 | 99.3 | 93.4 | 99.3 | 99.3 |
| MAC | 98.9 | 97.1 | 99.5 | 99.1 | 99.5 | 99.5 |
| GQA | 97.7 | 94.9 | 98.3 | 97.0 | 99.2 | 99.2 |

Table 4: Test results on CLEVR, demonstrating high accuracy at complex reasoning. GQA is the first approach to achieve high performance on both CLEVR and broad coverage QA tasks.

ing MAC cells (Hudson & Manning, 2018) into our decoder or FiLM layers (Perez et al., 2018) into our encoder would be straightforward, and may improve results, but we avoid these techniques to emphasise that generative decoding alone allows multihop reasoning.

Figure 4 shows an example of how the model decomposes the reasoning over the question. It initially pays attention to all shapes, but updates its attention mask after new words are read.

### 3.3 LEARNING FROM BIASED DATA

Many popular QA datasets are well known to contain biases that models can exploit. Examples include *when* questions paired with paragraphs that contain a single date. Models can exploit biases by learning simple heuristics such as selecting answers based on the expected answer type (Weissenborn et al., 2017; Rondeau & Hazen, 2018). Recent work has attempted to remove some biases (Goyal et al., 2017; Rajpurkar et al., 2018); we instead attempt to make training robust to bias.

We create deliberately biased training subsets of SQUAD based on named entity types: numbers, dates, and people. To construct each training set, we select questions whose answer is one of these types, but that type only appears once in the document (e.g. Figure 1b). The validation set is created from questions whose answers are the named entity type, but there must be multiple occurrences of that type in the document. Each training and validation set contains rougly 1000 questions.

We compare our model to two strong discriminative models, BiDAF (Seo et al., 2016) and QANet[3] (Yu et al., 2018). We also report three question agnostic baselines: a random answer of the correct type, the first answer of the correct type, and the GQA answer prior distribtion.

Results are shown in Table 5, and show that discriminatively trained models perform similarly to question-agnostic baselines. In contrast, our generative model learns to generalise meaningfully even from highly biased data, because it is trained to explain the whole question, not simply to answer it—demonstrating that on some QA tasks, there are clear advantages to generative modelling.

### 3.4 ADVERSARIAL EVALUATION

We evaluate on an adversarial version of the SQUAD dataset (Jia & Liang, 2017), which was created by adding a distractor sentence to each paragraph that can almost answer the question.

---

[3]We use the re-implementation from `https://github.com/NLPLearn/QANet`

| Single Model | Numbers | | Dates | | People | |
|---|---|---|---|---|---|---|
| | EM | F1 | EM | F1 | EM | F1 |
| Random Selection | 19.37 | 26.18 | 19.37 | 26.18 | 19.37 | 26.18 |
| First Occurrence | 29.36 | 35.11 | 34.64 | 42.08 | 26.38 | 32.26 |
| BiDAF | 33.02 | 42.14 | 35.41 | 43.83 | 30.05 | 37.28 |
| QANet | 31.99 | 40.58 | 39.98 | 47.82 | 30.26 | 38.56 |
| GQA (answer prior only) | 37.15 | 45.54 | 35.55 | 43.85 | 32.56 | 38.79 |
| GQA | 58.49 | 67.56 | 64.71 | 72.51 | 53.09 | 61.93 |

Table 5: Exact Match (EM) and F1 on biased subsets of SQUAD. All answers in each subset have the indicated named-entity type; training documents have only one answer with this type, but for testing there are multiple plausible answers. Discriminative models perform comparably to question-agnostic baselines, whereas our generative model learns to generalise.

| Single Model | ADDSENT | ADDONESENT |
|---|---|---|
| BiDAF (Seo et al., 2016) | 34.3 | 45.7 |
| RaSOR (Lee et al., 2016) | 39.5 | 49.5 |
| MPCM (Wang et al., 2016) | 40.3 | 50.0 |
| ReasoNet (Shen et al., 2017) | 39.4 | 50.3 |
| Reinforced Mnemonic Reader (Hu et al., 2018) | 46.6 | 56.0 |
| QANet (Yu et al., 2018) | 45.2 | 55.7 |
| GQA | 47.3 | 57.8 |

Table 6: F1 scores on ADVERSARIALSQUAD (from September 2018), which demonstrate that our generative QA model is substantially more robust to this adversary than previous work, likely because the additional adversarial context sentence cannot explain all the question words.

Table 6 shows that GQA outperforms the best previous work by up to 2.1 F1, making it the most robust model to these adversarial attacks. The improvement may be due to the model's attempt to explain all question words, some of which may be unlikely under the distractor (Figure 2).

### 3.5 LONG CONTEXT QUESTION ANSWERING

Finally, we extend our generative model to answering questions in a more challenging, multi-paragraph setting. While we train on single paragraphs, our model can be used to answer questions with multi-paragraph context. During training, our model $p(q \mid a, c)$ depends only on the content of the paragraph $c$ containing the correct answer $a$. In contrast, discriminative models need to be trained to discriminate against all negative answers from all paragraphs.

We use the multi-paragraph SQUAD dataset of Raison et al. (2018), where each question is paired with the entire corresponding Wikipedia article. For each question, we calculate $p(q \mid a)s(a, c)$ for the proposed answer span $a$ produced by the model, where $s(a, c)$ is the logits from the answer prior classifier. The maximum value across all paragraphs of that article is selected as the answer for that question. Table 7 shows that GQA outperforms previous work on this task by 2.5 F1. Further, discriminative models such as DrQA and Weaver require training in the multi-paragraph setting to perform well, which is expensive and may not scale to longer contexts. However, our generative approach performs well in the multi-paragraph test setting but only requires single paragraph training.

## 4 RELATED WORK

Our model is inspired by the classic noisy channel translation models of Brown et al. (1993), more recently explored by Yu et al. (2016), which were motivated by the ease of incorporating a prior over outputs. Generative models have been widely used in other language classification tasks, such as sequence tagging (Brants, 2000) and parsing (Collins, 1997; Dyer et al., 2016). Generative classification models became less popular because of the difficulty of modelling the input (Sutton & McCallum, 2012), a challenge we embrace as an additional learning signal. Recent work has shown

| Single Model | EM | F1 |
|---|---|---|
| DrQA* trained on paragraph | 59.1 | 67.0 |
| Weaver trained on paragraph | 60.6 | 69.7 |
| DrQA* trained on documents | 64.7 | 73.2 |
| Weaver trained on documents | 67.0 | 75.9 |
| GQA trained on paragraph | 71.4 | 78.4 |

Table 7: F1 scores on full document evaluation for SQUAD, which show our generative QA model is capable of selecting the correct paragraph for question answering even when presented with other similar paragraphs. Baselines are from (Raison et al., 2018).

the effectiveness of generative pre-training on unlabelled data (Peters et al., 2018; Radford et al., 2018), we show additional gains from training generatively on labelled data.

Several studies have explored the relationship between question answering and question generation. Duan et al. (2017) and Tang et al. (2017) train answering and generation models with separate parameters, but add a regularisation term that encourages the models to be consistent. They focus on answer sentence selection, so performance cannot easily be compared with our work. Tang et al. (2018) use question generation to provide an additional loss to improve question answering systems using GAN-like training. Li et al. (2017) apply similar techniques to visual QA. Our work differs in training a single model for the joint distribution of questions and answers, which can be used to calculate conditional distributions for question generation or answering. Sachan & Xing (2018) improve performance by generating new question-answer pairs for training from unlabelled text, which would be a possible extension to our work. Echihabi & Marcu (2003) describe an earlier method for answering questions in terms of the distribution of questions given answers—one conceptual difference is that their approach does not include a prior over answers.

Question generation has also been studied as a task in its own right. Heilman & Smith (2010) use a rule-based system to generate candidate questions, followed by statistical ranking. Du et al. (2017) use a sequence-to-sequence model that encodes paragraph and sentence level information to generate questions. Cardie & Du (2018) propose using a coreference mechanism to incorporate contextual information from multiple sentences. Liu et al. (2017) explore question generation from images, which they refer to as Inverse Visual QA. Yuan et al. (2017) fine-tune a question generation model using reinforcement learning, based on fluency and whether it can be answered. Although we train a question generation model, our focus is on using it to answer questions.

## 5    CONCLUSION

We introduced a generative model for question answering, which leverages the greater amount of information in questions than answers to achieve high performance in both language comprehension and reasoning. The approach demonstrates better robustness to biased training data and adversarial testing data than state-of-the-art discriminative models. There are numerous interesting directions for future work, such as combining information about an entity from multiple sources to generate questions. Given the rapid progress made on discriminative QA models in recent years, we believe there is significant potential for further improvements in generative question answering.

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

## A  TRAINING DETAILS

### A.1  SQUAD MODEL

**Pre-processing**  Questions and answers were tokenized with a simple rule-based tokenizer. We ignored training examples whose answers did not correspond to words in our tokenization. We also found it helpful to append the article title as an additional sentence at the end of paragraphs, as frequently question words make use of an entity mentioned in the title that is not in the paragraph. Finally, we replace question words with similar words from the context (based on sharing word stems or low edit distance), which makes it easier for the model to explain rare words and typographical errors in questions by using the pointer mechanism.

**Architecture**  The encoder contains 2 answer-independent LSTM layers and 3 answer-dependent LSTM layers, all of hidden size 128. The decoder contains 9 blocks, all with hidden size $d = 256$. We apply dropout ($p = 0.55$) to contextualised word representations, after encoder LSTM layers and after each decoder block (before residual connects). We also used word level dropout after contextualised embeddings for each encoder ($p = 0.1$) and decoder word ($p = 0.25$), and disallow use of the pointer mechanism with $p = 0.25$. All dropout masks are fixed across time-steps (Gal & Ghahramani, 2016).

**Optimisation**  We train generatively with batches of 10 documents, using a cosine learning rate schedule with a period of 1 epoch, warming up over the first 5 epochs to a maximum learning rate of $10^{-4}$. During fine-tuning, we freeze the answer-independent context encoding and $p(a|c)$ model, which both reduces memory requirements and makes learning more stable. If the correct answer is not in the beam, we make no update. Fine tuning uses stochastic gradient descent with single question batches, learning rate $5 * 10^{-5}$, and momentum $0.97$.

### A.2  CLEVR MODEL

**Architecture**  All hidden layers in the encoder have size 128. We use 3 blocks of convolution, batch normalisation, and ReLU activations. The first block uses a 1x1 convolution to project the pre-trained features to size 128, and the other blocks use 3x3 convolutions. We apply dropout with rate $0.1$ to the pre-trained image features. In the decoder we use a dimension $d = 256$ with 6 decoder blocks, with dropout ($p = 0.25$) before residual connections.

**Optimisation**  We optimise with stochastic gradient descent with momentum $0.9$, with an initial learning rate of $0.025$ which is decayed by a factor of 5 when there is no improvement for 10 epochs. For generative training we use batch size 1024. For fine tuning, we use initial learning rate $0.001$ and batch size 32.

