# OpenReview forum: "Generative Question Answering: Learning to Answer the Whole Question"
_ICLR.cc/2019/Conference_

### Official Review · AnonReviewer2 · 2018-10-30
**Great idea, well executed, well written**

**Rating:** 8
**Confidence:** 4

**Review:**

This paper introduces a generative model for question answering.  Instead of modeling p(a|q,c), the authors propose to model p(q,a|c), factorized as p(a|c) * p(q|a,c).  This is a great idea, it was executed very well, and the paper is very well written.  I'm glad to see this idea implemented and working.

Reactions:
- Section 2.1: Is there a bias problem here, where you're only ever training with the correct answer?  Oh, I see you covered that in section 2.6.  Great.
- Section 2.4: what happens when there are multiple QA pairs per paragraph or image?  Are you just getting conflicting gradients at different batches, so you'll end up somewhere in the middle of the two answers?  Could you do better here?
- Section 2.6: The equation you're optimizing there reduces to -log p(a|q,c), which is exactly the loss function used by typical models.  You should note that here.  It's a little surprising (and interesting) that training on this loss function does so poorly compared to the generative training.  This is because of how you've factorized the distributions, so the model isn't as strong a discriminator as it could be, yes?
- Section 3.1 (and section 2.6): Can you back up your claim of "modeling more complex dependencies" in the generative case?  Is that really what's going on?  How can we know?  What does "modeling more complex dependencies" even mean?  I don't think these statements really add anything currently, as they are largely vacuous without some more description and analysis.
- Section 3.3: Your goal here seems similar to the goal of Clark and Gardner (2018), trying to correctly calibrate confidence scores in the face of SQuAD-like data, and similar to the goals of adding unanswerable questions in SQuAD 2.0.  I know that what you're doing isn't directly comparable to either of those, but some discussion of the options here for addressing this bias, and whether your approach is better, could be interesting.

Clarity issues:
- Bottom of page 2, "sum with a vector of size d" - it's not clear to me what this means.
- Top of page 3, "Answer Encoder", something is off with the sentence "For each word representation"
- Section 2.5, "we first embed words independently of the question" - did you mean "of the _context_"?
- Section 2.5.2 - it's not clear to me how that particular bias mechanism "allows the model to easily filter out parts of the context which are irrelevant to the question".  The bias mechanism is independent of the question.
- Section 2.7 - when you said "beam search", I was expecting a beam over the question words, or something.  I suppose a two-step beam search is still a beam search, it just conjured the wrong image for me, and I wonder if there's another way you can describe it that better evokes what you're actually doing.
- Section 3.1 - "and are results..." - missing "competitive with"?
- Last sentence: "we believe their is" -> "we believe there is"

---

> ### Author Response · Authors · 2018-11-24
> **Response**
>
> Thanks for the helpful comments and feedback, which will let us improve the final version.
>
> - Section 2.4: what happens when there are multiple QA pairs per paragraph or image?  Are you just getting conflicting gradients at different batches, so you'll end up somewhere in the middle of the two answers?  Could you do better here?
>
> We don't think there's a problem here: each QA pair can be viewed as a sample from the space of QAs that can be asked for that context, and the model will learn to capture this distribution.
>
> > - Section 2.6: The equation you're optimizing there reduces to -log p(a|q,c), which is exactly the loss function used by typical models.  You should note that here.  It's a little surprising (and interesting) that training on this loss function does so poorly compared to the generative training.  This is because of how you've factorized the distributions, so the model isn't as strong a discriminator as it could be, yes?
>
> We believe the issue here is because even though the loss function is mathematically equivalent, our factorization requires us to learn a conditional language model, and the discriminative loss function does not provide enough learning signal to train such a model.
>
> > - Section 3.1 (and section 2.6): Can you back up your claim of "modeling more complex dependencies" in the generative case?  Is that really what's going on?  How can we know?  What does "modeling more complex dependencies" even mean?  I don't think these statements really add anything currently, as they are largely vacuous without some more description and analysis.
>
> Thanks for the feedback. The intuition behind this statement is that a generative model has to learn more information connecting the question and document, because generating a question is more difficult than answering it. We agree that it is unclear as currently written, and will rephrase or remove it.

---

### Official Review · AnonReviewer3 · 2018-11-02
**Interesting ideas but weak experiment results**

**Rating:** 6
**Confidence:** 4

**Review:**

In this paper, authors proposed a generative QA model, which optimizes jointly the distribution of questions and answering given a document/context. More specifically, it is decomposed into two components: the distributions of answers given a document, which is modeled by a single layer neural network; and the distribution of questions given an answer and document, which is modeled by a seq2seq model with a copy mechanism. During inference, it firstly extracts the most likely answer candidates, then evaluates the questions conditioned on the answer candidates and document and finally returns the answer with the max joint score from two aforementioned components.


Pros:
The paper is well written and easy to follow.

The ideas are also very interesting.

It gives a good ablation study and shows importance of each component in the proposed model.


Cons:
The empirical results are not good. For example, on the SQuAD dataset, since the proposed model also used ELMo (the large pre-trained contextualized embedding), cross attentions and self-attentions, it should be close or better than the baseline BiDAF + Self Attention + ELMo. However, the proposed model is significantly worse than the baseline (83.7 vs 85.6 in terms of F1 score). From my experience of the baseline BiDAF + Self Attention + ELMo, it obtains 1 more point gain if you fine tune the models.  On CLEVER dataset, I agree that incorporating with MAC cells will help the performance.

In Table 1, it should be clear if the authors could category those models into with/without ELMo for easy compassion. Furthermore, it is unclear how the authors select those baselines since there are many results on the SQuAD leaderboard. For example, there are many published systems outperformed e.g., RaSOR.

Questions:
During inference, generating answer candidates should be important. How the number of candidate affects the results and the inference time?

In SQuAD dataset, its answers often contain one or two tokens/words. What is the performance if removed length of answer feature?

During the fine turning step, have you tried other number of candidates?

---

> ### Author Response · Authors · 2018-11-24
> **Response**
>
> Thanks for the review and constructive comments!
>
> The main concern is that our results on SQuaD are beneath the current state of the art. Our generative approach means that our architecture is very different from existing models - we can't simply change the loss function for BiDAF, for example. These discriminative architectures have been carefully iterated on by a large community for several years. The fact that we are within a few points with the first generative approach is encouraging, and it seems reasonable that significant improvements would be possible with more development.
>
> We also emphasize that our model outperforms all discriminative models on adversarial SQuAD, demonstrating that it has learnt something more robust. We also show that our architecture can perform multi-hop reasoning, which has not been shown for any other strong SQuAD model.
>
>
> > In Table 1, it should be clear if the authors could category those models into with/without ELMo for easy compassion. Furthermore, it is unclear how the authors select those baselines since there are many results on the SQuAD leaderboard. For example, there are many published systems outperformed e.g., RaSOR.
>
> To keep the results table to a manageable size, we included a representative sample of existing approaches.
>
> > During inference, generating answer candidates should be important. How the number of candidate affects the results and the inference time?
> The inference time grows linearly in the number of answer candidates. We found that the beam starts to saturate at about 100 answers, covering about 99% of correct answers.
>
> Model                   		  	       EM / F1                      Inf Speed for Valid (sec)
> GQA, 250 answer candidates                  76.8 / 83.7			         617.69
> 200 answer candidates		      76.6 / 83.4		                     535.35
> 100 answer candidates		      76.2 / 83.1			         359.67
> 50 answer candidates			      74.6 / 81.4			         262.42
> 10 answer candidates			      55.7 / 61.4			         200.91
>
>
> > In SQuAD dataset, its answers often contain one or two tokens/words. What is the performance if removed length of answer feature?
>
> Model                   		  Generative (EM)                +Fine Tuning (EM / F1)
> GQA                        			  72.3                               76.8 / 83.7
> No answer length feature                     72.0			     73.8 / 80.1
>
> It is quite interesting that the length feature is particularly helpful for fine-tuning. During generative training, the question generation model is mostly exposed to short answers, because it is only shown the gold answer. However, at test time, it mostly sees very long answers, because most possible answers are long, and its performance may be weak on these. The answer length feature makes it easy for fine-tuning to compensate for this imbalance. We will update the paper with this ablation.

---

### Official Review · AnonReviewer1 · 2018-11-03
**Good ideas, more clarification about results/relevant work is necessary**

**Rating:** 7
**Confidence:** 4

**Review:**

This paper proposes a generative approach to textual QA on SQUAD and visual QA on CLEVR dataset, where, a joint distribution over the question and answer space, given the context (image or Wikipedia paragraphs) is learned (p(q,a|c)). During inference the answer is selected by argmax p(q,a|c) that is equal to p(a|c,q) if the question is given. Authors propose an architecture shown in Fig. 3 of the paper, where generation of each question word is condition on the corresponding answer, context and all the previous words generated in the question so far. The results compared to discriminative models are worse on SQUAD and CLEVR. Nevertheless, authors show that given the nature of the model that captures more complex relationships, the proposed model performs better than other models on a subset of SQUAD that they have created based on answer type (number/date/people), and also on adversarial SQUAD.

Comments / questions:

The paper is well written, except for a few parts mentioned below, all the equations / components are explained clearly. The motivation of the paper is clearly stated as using generative modelling in (V)QA to overcome biases in these systems, e.g., answering questions by just using word matching and ignoring the context (context=image or Wikipedia paragraph). I have the following questions / comments about the paper which addressing them by authors will help to better understand/evaluate the paper:
1.	In page 3 on the top of section 2.3, can authors provide a more clear explanation of the additional 32-dimensional embedding added to each word representation? Also in Table 2, please add an ablation how much gain are you getting from this?
2.	In the same page (page 3), section 2.4, paragraph 2, put the equation in a separate line and number it + clearly explain how you have calculated s^{endpoints} and s{length}.
3.	In page 4 section 2.5.2 paragraph 2, the way the bias term is calculated and the incentive behind it is not clear. Can authors elaborate on this?
4.	In page 6 section 3.2 the first paragraph authors claim that their model is performing multihop reasoning on CLEVR, while there is no explicit component in their model to perform multiple rounds of reasoning. Can authors clarify their statement?
5.	In section 3.3 the third paragraph, where authors explain the question agnostic baselines, can they clarify what they mean by “the first answer of the correct type”?
6.	In Table 5 and section 3.4 the second paragraph, authors are stating that “… The improvement may be due to the model’s attempt to explain all question words, some of which may be unlikely under the distractor”. It is very important that the authors do a complete ablation study similar to that of Table 2 to clarify how much gain is achieved using each component of generative model.
7.	In page 8 under related works:
a.	In paragraph 2 where authors state “Duan et al. (2017) and Tang et al. (2017) train answering and generation models with separate parameters, but add a regularisation term that encourages the models to be consistent. They focus on answer sentence selection, so performance cannot easily be compared with our work.”. I do not agree that the performance can not be compared, it is easily comparable by labeling a sentence containing the answer interval as the answer sentence. Can authors provide comparison of their work with that of Duan et al. (2017) and Tang et al. (2017)?
b.	In the same paragraph as 7.a, the authors have briefly mentioned “Echihabi & Marcu (2003) describe an earlier method for answering questions in terms of the distribution of questions given answers.” Can they provide a more clear explanation of this work and its relation to / difference with their work?

////////////
I would like to thank authors for providing detailed answers to my questions. After reading their feedback, I am now willing to change my score to accept.

---

> ### Author Response · Authors · 2018-11-24
> **Response**
>
> Thanks for the review and detailed feedback, which we’ll be happy to address in the final submission. Answers to questions are beneath.
>
> > In page 3 on the top of section 2.3, can authors provide a more clear explanation of the additional 32-dimensional embedding added to each word representation? Also in Table 2, please add an ablation how much gain are you getting from this?
>
> Thanks, have expanded the explanation, and an ablation is beneath. Including this embedding makes it easier for the model to learn the relationship between the document word and answer. However, it does significantly increase the computational cost of inference with the model, because we have to compute a separate contextualized document representation for each candidate answer.
>
> Model                   		  Generative (EM)                      Fine Tuning (EM / F1)
> GQA                        			  72.3                                       76.8 / 83.7
> No 32 dim embedding                          66.9				 69.2 / 75.9
>
>
> >	In page 6 section 3.2 the first paragraph authors claim that their model is performing multihop reasoning on CLEVR, while there is no explicit component in their model to perform multiple rounds of reasoning. Can authors clarify their statement?
> The model can perform multiple rounds of reasoning as a by-product of explaining the question word-by-word. On CLEVR, the model must explain each question word in turn, allowing it to track the relevant objects in complex chains of reasoning (see Figure 4).
>
> >	In page 4 section 2.5.2 paragraph 2, the way the bias term is calculated and the incentive behind it is not clear. Can authors elaborate on this?
> The motivation is that it can allow the model to quickly focus its attention on relevant parts of the paragraph (which is typically several hundred words). We found that it improved convergence pruning out irrelevant parts of the input.
>
> > 	In section 3.3 the third paragraph, where authors explain the question agnostic baselines, can they clarify what they mean by “the first answer of the correct type”?
> Here, we simply mean that e.g. for a question whose answer is a person, we return the first person in the evidence paragraph. We will clarify this in the paper.
>
> >	In Table 5 and section 3.4 the second paragraph, authors are stating that “… The improvement may be due to the model’s attempt to explain all question words, some of which may be unlikely under the distractor”. It is very important that the authors do a complete ablation study similar to that of Table 2 to clarify how much gain is achieved using each component of generative model.
>
> We chose not to present ablations for adversarial SQuAD in the submission, because there is no validation data, so we only performed a single run on the data with our best model (selected on the standard SQuAD data). The fact that our approach performs better than models that outperform it on SQuAD is strong evidence that it has learnt something more robust from the same training data. Please let us know if you still think including these ablations would be helpful.
>
> > a.	In paragraph 2 where authors state “Duan et al. (2017) and Tang et al. (2017) train answering and generation models with separate parameters, but add a regularisation term that encourages the models to be consistent. They focus on answer sentence selection, so performance cannot easily be compared with our work.”. I do not agree that the performance can not be compared, it is easily comparable by labeling a sentence containing the answer interval as the answer sentence. Can authors provide comparison of their work with that of Duan et al. (2017) and Tang et al. (2017)?
> There isn't actually enough detail in these papers to replicate their non-standard experimental setup---given that there are only a few sentences in each paragraph, a lot would depend on how exactly they segmented the input into sentences. However, the reported accuracies are only a few percentage points higher than our approach on this much easier task, so it seems unlikely that their results would be competitive.
>
> > b.	In the same paragraph as 7.a, the authors have briefly mentioned “Echihabi & Marcu (2003) describe an earlier method for answering questions in terms of the distribution of questions given answers.” Can they provide a more clear explanation of this work and its relation to / difference with their work?
> Echihabi & Marcu train a model for p(q|a,c) using a rather complex combination of heuristics and classical machine translation methods, and return the answer maximizing this distribution. A conceptual difference is that our approach models p(q,a|c), and to our knowledge is the first generative question answering model.  Beyond the fact that they use a model of p(q|a) for question answering, there isn't much overlap in terms of motivation or techniques.

---

### Author Response · Authors · 2018-10-24
**SQuAD Test Results**

Our SQuAD test results were missing from the submission because of a technical problem with the evaluation server. Our results are now available as 77.090 (Exact Match) 83.931 (F1). This model was submitted before the ICLR deadline,.

---

### Comment · AnonReviewer2 · 2018-11-15
**This paper should be accepted**

Just starting a conversation with other reviewers.  I feel pretty strongly that this paper should be accepted.  We should not be fixating on leaderboard performance numbers and blackbox comparisons.  Science is much more broad than "who has the best experimental result".  The presented method in the paper works well, it's a very interesting, novel idea, and the paper is well written.

---

### Author Response · Authors · 2018-11-24
**New experiment on long-context question answering**

Thanks to all the reviewers, we are happy that they all found the ideas in the paper to be interesting.

We’ve added one additional experiment beyond what was requested (Section 3.5). We explore question answering when the answer can be contained in one of many paragraphs. This task is computationally expensive to train properly with discriminative models, because at training time you would ideally want to discriminate against all the negative answers from all paragraphs. In our approach, most of the work is done by the model of p(q|a,c), which only depends the paragraph c containing the gold answer a. That means we can train the model using single paragraphs, but test it on multiple paragraphs.

We outperform the best previous work that was trained on single paragraphs (as ours was) by almost 10 F1, and the best approaches trained on multiple paragraphs by 2.5 F1. This experiment highlights a further advantage of generative question answering.

---

> ### Comment · AnonReviewer2 · 2018-11-25
> **Nice experiment, but missing related work**
>
> When I saw this description, I thought you were comparing against Clark and Gardner 2018 (https://arxiv.org/abs/1710.10723; DocQA).  I hadn't seen Weaver before, and I was surprised there there hasn't been a comparison between Weaver and DocQA (so I'm not actually sure which is better).  DocQA only requires training with two paragraphs at a time, not the full document, so the argument about scalable training rings a bit hollow (it's a constant factor, not dependent on document length).  It'd be best to compare against that work also if you want to make claims about multi-paragraph performance, or anything really on TriviaQA (looks like DocQA has also long since been beaten).

---

### Meta-Review · Area_Chair1 · 2018-12-13
**Clear accept ratings from reviewers.**

**Confidence:** 4
**Recommendation:** Accept (Poster)

**Metareview:**

All reviewers recommend accept.
Discussion can be consulted below.